# Chronic Chemogenetic Stimulation of the Nucleus Accumbens Produces Lasting Reductions in Binge Drinking and Ameliorates Alcohol-Related Morphological and Transcriptional Changes

**DOI:** 10.3390/brainsci10020109

**Published:** 2020-02-18

**Authors:** Dar’ya Y. Pozhidayeva, Sean P. Farris, Calla M. Goeke, Evan J. Firsick, Kayla G. Townsley, Marina Guizzetti, Angela R. Ozburn

**Affiliations:** 1Department of Behavioral Neuroscience, Oregon Health & Science University, Portland, OR 97239, USA; pozhiday@ohsu.edu (D.Y.P.); calliegoeke@gmail.com (C.M.G.); kaala96@gmail.com (K.G.T.); guizzett@ohsu.edu (M.G.); 2Research & Development, VA Portland Health Care System, Portland, OR 97239, USA; firsick@ohsu.edu; 3Chemistry Department, Portland State University, Portland, OR 97207, USA; 4College of Natural Sciences, Waggoner Center for Alcohol and Addiction Research, University of Texas at Austin, Austin TX 78712, USA; spfarris@utexas.edu

**Keywords:** binge drinking, genes, signaling pathways, DREADDs or chemogenetics, high drinking in the dark mice

## Abstract

Binge drinking is a dangerous pattern of behavior. We tested whether chronically manipulating nucleus accumbens (NAc) activity (via clozapine-N-oxide (CNO) and Designer Receptors Exclusively Activated by Designer Drugs (DREADD)) could produce lasting effects on ethanol binge-like drinking in mice selectively bred to drink to intoxication. We found chronically increasing NAc activity (4 weeks, via CNO and the excitatory DREADD, hM3Dq) decreased binge-like drinking, but did not observe CNO-induced changes in drinking with the inhibitory DREADD, hM4Di. The CNO/hM3Dq-induced reduction in ethanol drinking persisted for at least one week, suggesting adaptive neuroplasticity via transcriptional and epigenetic mechanisms. Therefore, we defined this plasticity at the morphological and transcriptomic levels. We found that chronic binge drinking (6 weeks) altered neuronal morphology in the NAc, an effect that was ameliorated with CNO/hM3Dq. Moreover, we detected significant changes in expression of several plasticity-related genes with binge drinking that were ameliorated with CNO treatment (e.g., *Hdac4*). Lastly, we found that LMK235, an HDAC4/5 inhibitor, reduced binge-like drinking. Thus, we were able to target specific molecular pathways using pharmacology to mimic the behavioral effects of DREADDs.

## 1. Introduction 

Binge drinking is a problematic pattern of behavior and often leads to the development of alcohol use disorders (AUD; https://niaaa.scienceblog.com/24/niaaa-scientists-provide-more-evidence-that-binge-drinking-may-indicate-vulnerability-to-alcohol-use-disorder/; 02/15/20). Binge drinking is defined by the National Institute on Alcohol Abuse and Alcoholism as having 4 to 5 drinks within 2 h and/or achieving a blood alcohol level (BAL) >80 mg% (https://www.niaaa.nih.gov/alcohol-health/overview-alcohol-consumption/moderate-binge-drinking; 02/15/20). Limited access ethanol drinking paradigms are used to model binge/intoxication in animal models (however, not all limited access paradigms and animal strains drink to intoxication). In the limited access Drinking in the Dark (DID) paradigm, mice are offered an ethanol solution early into the active period of their circadian cycle and can achieve BALs >80 mg%, suggesting they drink to intoxication [1]. C57BL/6J mice are typically reported as high drinking and can achieve BALs >80mg% in the DID paradigm; however, significant inbred strain differences have been observed suggesting there exists a genetic contribution to this phenotype [2,3]. The DID assay was used to independently create two lines of mice, HDID-1 and HDID-2, that were selectively bred (from genetically heterogeneous HS/Npt progenitors) for high BALs after DID [4,5]. HDID and HS/Npt mice have been extensively characterized and HDID mice represent a unique genetic animal model of risk for drinking to intoxication [6,7,8,9,10,11,12,13,14]. FDA approved compounds for treatment of AUD, as well as several investigational compounds, have been tested for efficacy in reducing binge-like drinking using the DID paradigm in C57BL/6J and HDID mice, as well as other genotypes, with mixed results [15,16,17,18,19,20]. Together, the results of these studies suggest that testing potential therapies in more than one inbred strain may represent a more beneficial strategy for clinical translation.

Koob and Volkow (2010) reviewed decades of clinical and pre-clinical studies to address the neural circuitry associated with the three stages of the addiction cycle: binge/intoxication, withdrawal/negative affect, and preoccupation/anticipation [21]. The nucleus accumbens (NAc) is identified as an important region for each of these three stages. There is also extensive evidence that altering activity in the NAc reduces alcohol drinking, craving, and relapse. To achieve this, clinical studies have applied deep brain stimulation (DBS) to the NAc in humans [22,23,24]. Human studies were carried out in males with treatment resistant alcohol use disorder (AUD), where NAc DBS of the shell relieved symptoms of craving and reduced relapse [22]. Pre-clinical studies have also found that DBS of the NAc reduced alcohol intake that was generalizable to different strains of rats and drinking paradigms [25,26]. Many of these studies report that DBS of either the NAc core or shell (or both) are effective in reducing drug-related behaviors. Although DBS of the NAc decreases alcohol drinking and craving, its effects are not lasting (reviewed in [27]). 

The promising results of NAc DBS studies led us to test whether DREADDs (Designer Receptors Exclusively Activated by Designer Drugs) could be used to alter neuronal activity in the NAc and whether they could alter binge-like drinking. DREADDs (developed and characterized by Dr. Bryan Roth and his research team) have been an important tool for neuroscience research, enabling identification of brain circuitry involved in specific behaviors [28]. Here, we tested the effects of clozapine-N-oxide (CNO) on binge drinking in mice expressing excitatory DREADDs (hM3Dq), inhibitory DREADDs (hM4Di), or control (GFP) in the nucleus accumbens core. We tested the effects of CNO/DREADDs in female mice from a high drinking mouse line (HDID-1; where >90% of mice drink to intoxication) and hypothesized that CNO/hM3Dq would decrease binge-like drinking. DREADDs are mutagenized G-protein coupled receptors (GPCRs), and since we know GPCRs play an important role in neuroplasticity, we also tested the hypothesis that chronic stimulation of DREADDs (4 weeks) could lead to lasting reductions in binge-like drinking. We hypothesized that this behavioral plasticity could be observed at cellular and molecular levels via lasting changes in neuronal morphology and the transcriptome. Here, we focused on identifying genes and biological pathways that were altered with chronic binge-drinking (6 weeks) and ameliorated with chronic CNO treatment (4 weeks). We then pharmacologically targeted these biological pathways in male and female HDID-1 mice to determine whether they reduced binge like drinking. 

## 2. Materials and Methods

### 2.1. Experimental Animals

Experimental Animals: Adult High Drinking in the Dark line 1 (HDID-1) mice (generously provided by Dr. John Crabbe, Portland, Oregon, USA) aged 2–5 months were used for all experiments in this study. HDID-1 mice are a genetic model of risk for drinking to intoxication and were selectively bred from a genetically heterogeneous stock, HS/Npt [4,5]. Mice were bred and maintained in a reversed 12/12-h light/dark cycle, with lights off at 08:30 and lights on at 20:30. Cages were made of polycarbonate (28 × 17 × 11.5 cm) containing about 2.5 cm of cob bedding. Purina brand food (5LOD, PMI Nutrition International, Brentwood, MO, USA) was suspended in a wire-top, and food and water were available ad libitum. Female mice were used in experiments 1 and 2 to facilitate comparison with data our group had previously published in C57BL/6J mice [29]. Male and female mice were used in experiment 3. All procedures in this study were performed in accordance with NIH Guidelines for the Care and Use of Laboratory Animals and were approved by the VA Portland Health Care System’s Institutional Animal Care and Use Committee.

### 2.2. Drinking in the Dark (DID)

Binge-like drinking of ethanol in mice was assessed using Drinking in the Dark (using a variation of Rhodes et al., 2005 as described below). All mice had daily access to 20% ethanol (*v*/*v*, in tap water; Decon Laboratories, Inc., PA, USA) for 2 h (offered three hours into lights off). 

### 2.3. Drugs

For experiments 1 and 2: 30 min prior to DID, mice received IP (intraperitoneal) injections of either VEH (vehicle: 1% DMSO (Hybri-Max, Sigma Life Sciences, MO, USA) in saline (0.9% NaCl; Baxter International, IL, USA)), or 1 mg/kg CNO (1% DMSO in saline; RTI International, NC, USA). For experiment 3, 60 min prior to DID, mice were injected IP with either vehicle (5% DMSO in Dulbecco’s phosphate buffered saline; Gibco, Sigma Life Sciences, MO, USA) or 5 or 20 mg/kg LMK 235 (Selleck Chemicals, LLC, TX, USA).

### 2.4. Experiment 1

Female HDID-1 mice (S32.G34) were anesthetized with a mixture of ketamine (125 mg/kg) and xylazine (12.5 mg/kg) in saline and received bilateral stereotaxic infusions of 1 μL purified high titer AAV into the NAc (rAAV2-hSyn-eGFP (GFP; *n* = 15; control), rAAV2-hSyn-HA-hM3D(Gq)-IRES-mCitrine (hM3Dq; *n* = 13; excitatory DREADD), or rAAV2-hSyn-HA-hM4D(Gi)-IRES-mCitrine (hM4Di; *n* = 8; inhibitory DREADD)). Injections were administered to the following coordinates targeting the NAc core (from bregma, in mm): angle 10°, AP +1.5, Lat +1.7, DV −4.6 and −4.0; 0.5 μL delivered at DV −4.6 and 0.5 μL delivered at DV −4.0. AAV titers were 2-3.5 x 10^12^ vg and were purchased from the University of North Carolina Viral Vector Core. Two weeks later, mice were individually housed and habituated to novel sipper tubes for one week. The DID experiment was then carried out daily for 6 weeks. Mice were serially treated with vehicle prior to DID during week 1 to establish baseline drinking, CNO during weeks 2–5 to measure the effects of chronic treatment, and then mice were treated with vehicle again during week 6 to determine if there were any lasting effects of chronic CNO treatment. The serial treatment is represented in the figures as VEH/CNO/VEH. Ethanol intake was measured as described in Purohit et al., 2018 [29]. At the completion of this study, mice were deeply anesthetized and transcardially perfused with 0.01 M PBS and 4% *w/v* paraformaldehyde (PFA). Brains were extracted, sectioned at 30 µm using a freezing stage microtome (Model No. 860, American Optical Corp., Buffalo, New York, NY, USA), and processed for immunofluorescence to verify injection placements. Immunohistochemical staining was performed following standard procedures using rabbit anti-GFP or anti-HA primary antibody (anti-GFP, 1:20,000 dilution: Catalog#ab290, AbCam, Cambridge, MA, USA; anti-HA, 1:1000 dilution: Catalog #3724S, Cell Signaling Technologies, Danvers, MA, USA) and goat anti-rabbit Alexa 488 secondary antibody (1:500 dilution: Catalog #A-11088, Thermofisher Scientific, Waltham, MA, USA). Brain sections were mounted on microscope slides (Catalog #12-550-15, Fisherbrand, Waltham, MA, USA) using Vectashield Antifade Mounting Medium with DAPI (Vector Laboratories, Burlingame, CA, USA) and observed using an Olympus BX60 fluorescence microscope (Olympus, Center Valley, PA, USA). Animals were excluded if expression was not localized to the NAc (hM4Di *n* = 2, GFP *n* = 3).

### 2.5. Experiment 2

Female HDID-1 mice (S35.G37) were stereotaxically injected with 0.5 uL rAAV2/5-CMV-Cre-GFP and 0.5 uL rAAV2-hSyn-DIO-hM3Dq-mCherry bilaterally into the NAc targeting the NAc core/shell border region (from bregma, in mm: angle 10°, AP +1.34, Lat +1.5, DV −4.5). AAV titers were 4-8 x 10^12^ vg and were purchased from the University of North Carolina Viral Vector Core (Cre) and Add Gene (DIO constructs). Two weeks later, mice were individually housed and habituated to novel sipper tubes for one week prior to carrying out a 6 week (7 days/week) DID experiment (as described for experiment (1). There were four groups of mice: (a) mice treated with vehicle that only had access to water (control group), (b) mice treated with VEH/CNO/VEH that only had access to water (to test the effects of CNO on neuronal morphology and the transcriptome), (c) mice treated with vehicle that had access to ethanol (to test the effects of chronic binge-like drinking on morphology and the transcriptome), (d) mice treated with VEH/CNO/VEH that had access to ethanol (to test the effects of CNO and chronic binge-like drinking on neuronal morphology and the transcriptome). Mice were serially treated with vehicle prior to DID during week 1 to establish baseline drinking, then vehicle or CNO during weeks 2–5 to measure the effects of chronic treatment, and then mice were treated with vehicle again during week 6 to determine whether there were any lasting effects of chronic CNO treatment. Ethanol intake was measured as described in Purohit at al., 2018 [29]. For neuronal morphology analysis, female HDID-1 mice (S34.G36; *n* = 4–5/fluid type/treatment) were deeply anesthetized 22 h after the last DID and transcardially perfused with 0.01 M PBS and 4% *w*/*v* paraformaldehyde (PFA). Brains were extracted and placed in 4% PFA for 24 hr at 4 °C before processing for Golgi-Cox staining as described in Bayram-Weston et al., 2016 [30]. After 14 days at room temperature and low ambient light, the tissue was transferred into a 30% *w*/*v* sucrose solution for 24–72 hr (low light at 4 °C), prior to sectioning and mounting. Brains were sectioned coronally using a Compresstome at 200 μm (Model VF 300-0Z Microtome with AutoZero Z technique, Precisionary, Greenville, NC, USA) and mounted on gelatin-coated slides. Slides were prepared according to Section 2.3 of Bayram-Weston et al., 2016 [30], and cover-slipped with Permount™ Mounting Medium (Fisher Scientific Co., Waltham, MA, USA). For transcriptomic analysis, mice (*n* = 11–12/fluid type/treatment) were euthanized 22 h after the last DID session via cervical dislocation and rapid decapitation. Whole brains were frozen on powdered dry ice, 300 μm coronal sections were collected using a cryostat, and 1.0 mm frozen tissue punches were collected from the NAc and processed for RNA Sequencing. 

Microscopy: NAc medium spiny neurons were traced blind to condition using the software Neurolucida (Version 11, MBF Bioscience, Williston, VT, USA) on a Leica DM500b microscope using a DFC36 FX camera. Parameters analyzed were: complexity (defined as (sum of the terminal orders + the number of terminals) × (total dendritic length/number of dendrites)), average dendrite length (µm), number of nodes, number of ends, total dendritic length (µm), sum of the terminal orders (defined as number of sister branches encountered when tracing a dendrite from the tip back to the cell body), dendrite quantity, and branch sum (defined as the total number of segments between nodes). Neuronal complexity is a measure automatically derived from Neurolucida. In addition, Sholl analysis was carried out. The parameters analyzed in each 10 µm radius were: number of intersections, length (µm), number of nodes, number of ends, and the area under the curve was also calculated. The number of animals and cells per group that were included in analyses are given in Table 1.

RNA Sequencing and Validation of viral gene transduction: NAc tissue punches were mechanically homogenized in PureZol and RNA was isolated using the Aurum Total RNA Fatty and Fibrous Tissue kit (Bio-Rad Laboratories, Inc., Hercules, CA, USA). RNA (100 ng) was processed to cDNA using the BioRad iScript cDNA synthesis kit (Catalog #: 1708890; Bio-Rad Laboratories, Inc., Hercules, CA, USA) according to the manufacturer’s protocol. Quantitative real-time PCR was used to measure expression of mCherry to confirm viral expression of hM3Dq-mCherry in samples. A PrimePCR custom assay for mCherry (FOR: 5′-AGCGCGTGATGAACTTCGA-3′ REV: 5′-CGCAGCTTCACCTTGTAGATGA-3′; PROBE: 5′-CCGTGACCCAGGACTC; 5′ 6-FAM, 3′ Iowa Black FQ) and Rps18 (Mouse, FAM, qMmuCEP0053856) probes were used (Bio-Rad Laboratories, Inc., Hercules, CA, USA). mCherry expression levels (relative to Rps18) were verified as detectable and similar for all samples (data not shown). RNA samples were sent to the Massively Parallel Sequencing Shared Resource (MPSSR) at Oregon Health & Science University for quality assessment (via 2100 BioAnalyzer, Agilent Technologies, Palo Alto, CA, USA), library preparation, and sequencing. The library was prepared using the Illumina TruSeq RNA-Seq Library Protocol. Sequencing was executed via an Illumina HiSeq 2500 Sequencer (Illumina, San Diego, CA, USA) with poly(A)+ stranded selection and paired-end reads at 50 cycles. 

### 2.6. Experiment 3

Male and female HDID-1 mice (S38.G40, S39.G41, S41.G43) were individually housed and habituated to novel sipper tubes one week prior to DID. Mice were tested four days per week (as in [1]). Mice were serially treated with vehicle prior to DID (during weeks 1 or 1 and 2) to establish baseline drinking, and then 0, 5, or 20 mg/kg LMK 235 over the next 2 weeks to measure the effects of chronic treatment with this HDAC4/5 inhibitor. Ethanol intake was measured using volumetric tubes in 2 h intervals on all 4 days [8].

### 2.7. Data Analysis and Statistics

#### 2.7.1. Behavior

For experiments 1 and 2, daily ethanol intake data was averaged for each mouse for each treatment period. Intake data were then analyzed using a repeated measures one-way analysis of variance (ANOVA) to determine whether there was an effect of treatment period on drinking. For significant results (*p* < 0.05), post-hoc tests were carried out to compare baseline drinking to clozapine-n-oxide (CNO) treatment and washout periods. For experiment 3, drinking data for each 2 h interval was averaged for each period (baseline vs. treatment weeks). Data was analyzed using a two-way repeated measures ANOVA (dose × treatment period).

#### 2.7.2. Morphology

As the data was nested, with multiple neurons within a brain (dependent) being compared to multiple neurons from other brains (independent), we used a linear mixed effect analysis [31]. We did this using R and the package lme4 to perform a multilevel analysis [32]. The animal was accounted for as a random effect, due to multiple cells for each animal being examined. We used the fluid type [ethanol (EtOH) vs. water (H_2_O)] and the treatment (CNO vs. Vehicle) as fixed effects, and *p*-values were obtained by likelihood ratio tests. For all morphometric parameters, we examined the main effect of both variables, as well as their interaction. We then corrected for multiple comparisons using the Benjamini–Hochberg approach to adjust the nominal p-values to the q-values of the False Discovery Rate (FDR) [33]. Sholl analyses were analyzed by radius as well as area under the curve (AUC). *p*-values and *q*-values less than 0.05 were considered significant, and data was presented as mean ± standard error of the mean (SEM).

#### 2.7.3. Transcriptomic Analysis

Raw FASTA files generated from sequencing were aligned to the Genome Reference Consortium Mouse Reference 38, based on the *Mus musculus* strain C57BL/6J (GRCm38; also known as mm10). Basecall quality was assessed using FastQC [34] and alignment was run using STAR: ultrafast universal RNA-seq aligner [35] (version: 2.6.0) where paired-end reads were specified, the maximum number of mismatches per pair was indicated as 3 and maximum number of multiple alignments allowed per read was indicated as 1. Count data was generated using HTSeq (version 0.9.1) against the Encode vM4 annotation with paired-end read order specification [36]. After filtering, normalization and differential expression were performed using Bioconductor (version 3.34.9) in R [37]. The Limma voom function was utilized to perform empirical Bayes-moderated t-statistics to identify Differentially Expressed Genes (DEGs, *p* < 0.05) between fluid and treatment groups [38]. Mean-variance relationships and observational-level weights were calculated from log2-counts per million (log-CPM). Gene expression overlap was identified using BioVenn [39]. Expression patterns for genes mediating morphology were based on comparisons made during morphological analysis. Groups compared included only EtOH(VEH) vs. H_2_O(VEH) and EtOH(CNO) vs. H_2_O(CNO). Target genes identified for comparison were based on literature by Russo et al., 2010, Arikkath et al., 2012 and Uys et al., 2016 [40,41,42]. Weighted gene co-expression network analysis (WGCNA; version 1.67) was performed using R implementation [43]. A signed, hybrid co-expression network was constructed covering 24,421 genes meeting the cut-off criterion of 1 CPM. Prior to network construction, 10% of genes exhibiting the lowest correlation values were removed via SUMCOR as in Tritchler et al., 2009 [44]. The co-expression network was constructed via Pearson correlation between all gene pairs and the exponentiation of the resulting absolute value to a power beta = 7. A consensus network was constructed utilizing samples from all sequenced RNA samples. Modules were detected in each network via hierarchical clustering where average linkage and the WGCNA cuttreeHybrid function was used with the following parameters: cutHeight = 0.998, minClusterSize = 100, and deepSplit = 0. Total network connectivity (degree) of each gene was calculated via the sum total of all edges in the network, where modular connectivity restricted the edges included to the gene’s own module. Pseudo and Riken genes exhibiting abnormally high connectivity were sequestered and removed. Module quality was assessed using the WGCNA modulePreservation function. The geneMania and GOrilla enrichment algorithms were used to identify gene ontology annotation and enrichment for biological pathways [45,46]. Ontology for transcription factors was performed using the enrichR algorithm [47]. All code used is accessible via https://github.com/Pozhidayevad/Ozburn_RNA-Seq_Analyses. RNA Seq data is accessible via GEO (accession number GSE).

## 3. Results

### 3.1. Experiment 1: Chronically Increasing Gq Signaling in the NAc Produces Lasting Reductions in Binge-Like Drinking 

To identify the effect of manipulating NAc activity (via CNO and hM3Dq- or hM4Di-DREADDs) on binge-like ethanol drinking behavior, HDID-1 mice were subjected to the DID paradigm. The effects of treatment (via serial administration of vehicle, CNO, and vehicle) on ethanol binge-like drinking in mice expressing GFP (Figure 1A), hM4Di (Figure 1B), and hM3Dq (Figure 1C) is presented as the mean (± SEM) ethanol intake (g/kg/2h) across each treatment period. Mice expressing GFP or hM4Di in the NAc did not exhibit significant changes in ethanol intake during CNO treatment or vehicle washout periods (relative to baseline intake). However, mice expressing hM3Dq in the NAc exhibited a significant reduction in ethanol intake during chronic CNO treatment and during the vehicle washout periods relative to vehicle baseline intake (one way ANOVA—F(2,35) = 4.84, *p* < 0.01; post-hoc—VEH(baseline) vs. CNO, *p* < 0.05; VEH (baseline) vs. VEH (washout), *p* < 0.05). Thus, chronically increasing Gq signaling in the NAc produced lasting effects on binge-like drinking behaviors.

### 3.2. Experiment 2: Chronically Increasing Gq Signaling in the NAc Produces Lasting Reductions in Binge-Like Drinking, the Transcriptome, and Neuronal Morphology

#### 3.2.1. Behavioral Results

Results from experiment 1 showed that chronic stimulation using hM3Dq excitatory DREADDs produced a robust and lasting reduction in binge-like drinking in a model of genetic risk for binge-like drinking, thus the previous study was replicated using hM3Dq DREADDs exclusively. Here, the results of chronic CNO administration in mice hM3Dq excitatory DREADDs in the NAc were similar to data from experiment 1. Figure 2A shows ethanol drinking in response to chronic treatment with vehicle throughout the duration of the experiment (open bar). The EtOH(VEH) group experienced no significant changes at any point of the experiment in average ethanol intake (g/kg/2h DID session) relative to baseline intake. Conversely, the group treated with CNO shown in Figure 2B experienced significant and lasting reductions in ethanol intake during CNO treatment (one way ANOVA—F (2,35) = 10.49, *p* < 0.01; Dunnett post-hoc test revealed significant differences for VEH baseline vs. CNO (*p* < 0.01) and VEH baseline vs. VEH washout (*p* < 0.05)). Replication of this finding provides strong evidence for behavioral plasticity. We next sought to identify how this plasticity manifested on a morphological and transcriptomic level.

#### 3.2.2. Morphological Results

Behavioral plasticity has morphological underpinnings, which can be observed and analyzed. Therefore, we measured medium spiny neurons in the NAc for a variety of morphometric parameters, in order to determine the effects of treatment (CNO or vehicle), fluid type (EtOH or H_2_O), or whether there was an interaction between the two treatments. We analyzed neuronal complexity, determined via the equation (sum of terminal orders + number of terminals) × (total dendritic length/number of dendrites) (Figure 3A), length per dendrite (Figure 3B), number of nodes (Figure 3C), number of ends (Figure 3D), total dendritic length per neuron (Figure 3E), the sum of terminal orders, which was defined as the number of sister branches encountered when tracing a dendrite from the tip back to the cell body (Figure 3F), the quantity of dendrites (Figure 3G), and the branch sum, defined as the total number of segments between nodes (Figure 3H). In most parameters, there was an increase following exposure to EtOH (statistics for analyses are listed in Appendix A). This main effect was significant for neuronal complexity, the sum of terminal orders, and the branch sum, approached significance following corrections for multiple comparisons in the length per dendrite and the number of ends, and was significant before corrections for multiple comparisons in the number of nodes (Appendix A). There were no significant changes in the total dendritic length or the number of dendrites. There was also no effect of the injection (CNO or Vehicle), nor was there an interaction between the injection and exposure conditions. However, it is worth noting that the effects observed for the EtOH (VEH) group were not observed in the EtOH (CNO) group. 

We also performed Sholl analyses for these same neurons to quantify neuronal dendritic complexity. We analyzed intersections (Figure 3I,J,K), length (Figure 3L–N), nodes (Figure 3O–Q) and ends (Figure 3R–T) by radius (10 µm), and examined each radius individually with multilevel analyses and corrections for multiple comparisons. Since the primary effect seen in the neuron summary was an effect of EtOH in the vehicle injection condition, this is the primary comparison analyzed. We found that there were sporadic significant differences in radii, with no overall pattern of effects in any variable (statistics for the analyses are listed in Appendix A). Therefore, we elected to also examine the area under the curve (AUC). In this analysis, we found a similar effect as we did with the morphometric parameter analysis, as there was a greater AUC in the EtOH exposed groups when compared with H_2_O exposed groups, an effect which was significant in the nodes (Figure 3Q) and ends (Figure 3T), and approached significance in intersections (Figure 3K) and length (Figure 3N). Also similar to the morphometric parameter analysis was that this difference between EtOH and H_2_O exposure was no longer seen in the CNO injection condition, although there was no interaction between the injection and exposure, or effect of the injection. 

#### 3.2.3. Transcriptomic Results

To identify gene expression changes related to binge drinking and CNO treatment, differential expression analysis was performed using four groups from experiment 2. These four groups were defined as untreated binge-drinking (EtOH (VEH)), treated binge-drinking EtOH (CNO), treated water drinking (H_2_O (CNO)) and control water drinking (H_2_O (VEH)) animals. Comparisons made between groups involved the pairwise comparison of all groups with the control H_2_O (VEH) only. In this way, we identified differentially expressed genes (DEGs) from all groups relative to the control (Appendix A). The lists of DEGs were then compared for either overlapping or unique genes based on each group (Appendix A). The overlap in these DEGs is illustrated in Figure 4A. Here, 688 (out of 1473) significant DEGs were uniquely differentially expressed in EtOH (VEH). In EtOH (CNO) 1431 (out of 2377) genes were uniquely differentially expressed and in H_2_O (CNO) 612 (out of the 1157) genes were differentially expressed. The overlap for DEGs was highest between the EtOH (VEH) and EtOH (CNO) (*n* = 513) and lowest between EtOH (VEH) and H_2_O (CNO) (*n*= 112). Expression patterns for the unique DEGs identified for EtOH (VEH), EtOH (CNO) and H_2_O (CNO) are shown in Figure 4B. Each row represents a group, while each column represents a gene transcript. Transcripts are color-coded based on log-2 fold change from positive (blue) to negative (red). Gray indicates minimal change in expression. To identify genes that were significantly changed with chronic binge drinking and ameliorated by CNO, we calculated the pairwise Euclidian distance in expression patterns (for the 688 unique DEGs identified for the EtOH (VEH) group) and present the data in Appendix A and discuss these findings in the Appendix A. 

To identify the transcriptomic mechanisms of morphological changes observed in previous analyses, we identified gene expression changes of key molecular mediators of structural plasticity (details and data tables are provided in the Appendix A and methods). The heatmap in Figure 4C illustrates the expression patterns of 57 genes previously associated with increased (or decreased) neuronal structure. Columns indicate the comparisons performed for EtOH (VEH) vs. H_2_O (VEH) or EtOH (CNO) vs. H_2_O (CNO). DEG lists and expression values are listed in Appendix A. The significance of each gene relative to its comparison within the column is indicated, as * *p* < 0.05, ** *p* < 0.01. Here, 11 DEGs were identified between the two comparisons: 8 were found to be differentially expressed in EtOH (VEH) vs. H_2_O (VEH) and 3 in EtOH (CNO) vs. H_2_O (CNO). No overlap was observed within the DEGs between the two comparisons. Of the DEGs identified, all were associated with increased structural changes, excluding *Dlg4* and *Mef2d.* The 8 DEGs identified within the EtOH (VEH) vs. H_2_O (VEH) comparison included *Nrtk2, Wasf2, Dlg4, Grin2b, Grin2a, Was, Actl6a and Mir132,* where *Nrtk2, Wasf2, Grin2b, Grin2a,* and *Was* were significantly upregulated and *Dlg4, Actl6a and Mir132* were downregulated. The 3 DEGs identified within the EtOH (CNO) vs. H_2_O (CNO) comparison included *Mef2d*, *Camk2g* and *Pak1*, where *Camk2g* and *Pak1* were significantly upregulated and *Mef2d* was downregulated. In conjunction with the morphological analyses presented, these results suggest that expression of the mediators of structural plasticity are altered with ethanol drinking and ameliorated with CNO treatment.

The results of gene ontology (GO) term enrichment for biological processes in ETOH (VEH) are highlighted in Figure 5A. GO annotation of the ranked DE genes in EtOH (VEH) revealed a significant enrichment in gene-associated processes such as macromolecule metabolism, protein metabolism, cellular macromolecule metabolism, positive regulation of cellular processes, nitrogen compound metabolism, regulation of macromolecule metabolism and cellular protein metabolism among others (Appendix A). Genes overrepresented in these categories are shown in the clustergram in Figure 5A, where enriched terms are rows and input genes are columns. Cell colors in the matrix indicate which gene is ranked first within each associated GO term. The color depth of the cell corresponds to the legend and indicates the p-value of the enrichment term. Dark red indicates high significance and light red indicates low. The top 9 genes ranked first in the top 20 significant GO terms were *Cpsf3l, Hdac4, Fgfr3, Xpo4, Emilin2, Kiss1r, Trak2, Camta1* and *Fam135b.* In particular, *Hdac4* was ranked first in 13 of the top 20 GO categories. Among all categories, histone deacetylase 4 (*Hdac4*), a class II HDAC that deacetylates lysine residues of core histones and represses transcription via MEF2C and MEF2D, tended to be most prevalent in those related to the regulation of metabolic cellular processes. Comparison of the total normalized gene expression of *Hdac4* between groups is presented in Figure 5B. We observed a significant difference in expression between H_2_O and EtOH groups receiving vehicle treatment (two-way ANOVA: Fluid × Treatment interaction—F(1,41) = 4.45, *p* < 0.05; main effect of fluid—F(1,41) = 6.23, *p* < 0.05; post-hoc test revealed EtOH(VEH) vs H_2_O(VEH), *p* < 0.05). 

WGCNA was performed to identify the impact of chronic binge-like drinking and the lasting effects of CNO treatment on coordinated regulated gene expression networks. Though not discussed here, the results implicated the direct susceptibility of gene expression in a specific module to binge-like drinking and highlighted it as a point of interest (results in Appendix A).

### 3.3. Experiment 3: Pharmacologically Testing the Role of the Overrepresented Gene, Hdac4, in Binge-Like Drinking 

The effects of HDAC4/5 inhibitor, LMK-235, on binge-like alcohol drinking were significant. Notably, reductions in binge drinking became more apparent over time, suggesting transcriptional mechanisms. When comparing ethanol intake across baseline and treatment weeks, a two-way ANOVA revealed a significant main effect of treatment period on ethanol intake (Figure 5C; F(2,152) = 5.16, *p* < 0.01). Previous work has shown that the HDAC4/5 inhibitor, LMK-235, requires several administrations to exert its effects on signaling and morphology. We followed up on this finding by calculating a change score for each animal (by subtracting the average intake during treatment week 1 or week 2 from the average intake during baseline; shown in Figure 5D). Two-way ANOVA (dose × treatment week) revealed a significant main effect of dose, as well as a significant main effect of treatment week (dose: F(2,76) = 3.60, *p* < 0.05; treatment week: F(1,76) = 2.44, *p* < 0.05).

## 4. Discussion

Despite the prevalence of AUD and associated binge-related alcohol abuse, individuals often receive limited or no treatment. Currently, three FDA-approved medications are available for AUD treatment: disulfiram, acamprosate and naltrexone [48]. However, these medications are often underutilized, only improve symptoms in a small fraction of treated individuals, and any improvements do not last beyond treatment [49]. Further, although DBS of the NAc appears to be a potentially promising therapeutic approach, it is quite invasive and does not produce lasting reductions in drinking or craving. Thus, it is clear there exists a need for therapeutic approaches that produce robust and lasting effects. Here, we have shown that chronic binge-drinking induces lasting changes in neuronal morphology and gene expression networks, which can be ameliorated via chronic stimulation of Gq signaling (via CNO/hM3Dq) in the NAc. Furthermore, we found that we could pharmacologically target HDAC4, a gene that is altered with chronic binge-like drinking and ameliorated with CNO/hM3Dq, to reduce binge drinking. 

### 4.1. Animal Models of Binge Drinking and DREADDs

We and others have previously used HDID-1 mice as a reliable and predictive genetic model for high intensity binge-like drinking. This is due to their ability to achieve pharmacologically relevant blood alcohol levels (BALs), exhibit behavioral impairment after binge-like drinking, and exhibit relapse and withdrawal behavior following binge-like drinking sessions [6]. Genetic selection for high BALs has not altered the preference of mice to other tastants, nor has it affected their ability to metabolize alcohol [3]. Herein, the DID paradigm was employed for its high-throughput, limited access approach which did not require the use of tastants to motivate excessive levels of ethanol consumption during the dark cycle. In our study, we observed robust reductions in ethanol intake of female HDID-1 mice expressing hM3Dq in the NAc when treated with CNO during a DID limited access paradigm. These effects were observed for the entirety of the CNO treatment period (28 days), suggesting mice expressing hM3Dq in the NAc did not develop tolerance or show sensitization to the effects of chronic CNO administration. Moreover, the decreased intake persisted for at least 7 days during the vehicle “washout” period. We did not observe any change in ethanol drinking for female mice expressing GFP or hM4Di in the NAc, suggesting our findings are specific to HDID-1 mice expressing hM3Dq. However, there may exist sex differences, or paradigm-specific effects, in the effects of CNO/DREADDs for the NAc. Cassataro et al. (2014) found that CNO reduced binge-like drinking in C57BL/6J males expressing hM4Di in the NAc [50]. Direct comparison of these studies should be cautioned due to differences in methodology (length of ethanol access was 2 h, CNO was administered IP, and only females were used in the current DREADD study, and length of ethanol access was 2 and 4 h, CNO was administered via tap water to male mice in [50]. We are currently following up on these differences by conducting studies with both males and females to evaluate the role of sex as a biological factor. Taken together, these results provide supportive evidence for chronic Gq stimulation of the NAc as a causal factor in molecular and behavioral plasticity associated with lasting reductions of binge-like drinking in female mice.

### 4.2. Effects of Binge-Drinking and DREADDs on NAc Neuronal Morphology 

We found that chronic binge-like drinking resulted in increased complexity of medium spiny neurons in the NAc of HDID-1 female mice, an effect that was ameliorated by chronic treatment with CNO. Interestingly and unexpectedly, no effect of CNO was observed in water drinking mice, suggesting that chronically increasing Gq signaling in the NAc does not produce robust and lasting changes in neuronal complexity. We were unable to identify any published reports on the effects of DID or binge-like drinking on neuronal morphology in the NAc (or any other brain region) for comparison. However, there are several published studies evaluating the effects of chronic intermittent ethanol vapor on neuronal morphology in different brain regions. Most relevant to this work are studies by Wang et al. (2015), Uys et al. (2016), and DePoy et al. (2013) evaluating the effects of chronic intermittent ethanol vapor (CIE) on aspects of neuronal morphology in the dorsal and ventral striatum [42,51,52]. In brief, these studies report that CIE results in increased spine density, increased dendritic diameter, increased dendritic length and number of processes. Moreover, alcohol-induced increases in neuronal complexity in the dorsal striatum are correlated with increased AMPA receptor activity specifically in the dopamine D1 receptor expressing MSNs (medium spiny neurons), suggesting that alcohol enhances the activity of D1 MSNs in the direct pathway [52]. Notably, striatal circuits associated with D1- and D2-expressing MSNs are not equivalent for the dorsal and ventral striatum [53]. The role of D1- and D2-expressing MSNs of the NAc core in binge-like drinking is currently under study in our laboratory. In summary, although binge-like drinking in DID is likely modeling different aspects of AUD than CIE, it appears that both alcohol drinking models result in morphological changes in the dorsal and ventral striatum that manifest as increased neuronal complexity. Although additional studies are needed to better understand the effects of alcohol on morphology in different species, strains, sexes, and drinking paradigms.

Neuroadaptative changes, such as increased complexity, in NAc MSNs induced by ethanol exposure are the products of dysregulation in signaling systems, gene transcription and protein expression at the cellular level [54]. Previously, it has been shown that both opiate and stimulant drugs of abuse disrupt numerous genes that encode cytoskeleton regulatory proteins as well as their respective transcriptional factors [40]. These include cell surface receptors, adhesion molecules, signaling molecules, scaffolding proteins, GTPases regulating the actin cytoskeleton and guanine nucleotide exchange factors (GEFs) activated by GTPases. These mediators are associated with directional control of neuron morphogenesis. For example, neurotrophic factor components such as CREB and MAPK have been shown to increase dendrite branching, while scaffolding proteins such as PSD95 have been shown to act as stop signals for proximal dendritic branching [41]. In our work, we observed a significant upregulation of genes associated with increased dendritic branching and complexity such as *Ntrk2* (encodes BDNF tyrosine receptor kinase-B (TRKB)), *Wasf2, Was* (genes encoding Wiskott-Aldrich Syndrome proteins), *Grin2a, and Grin2b (*genes encoding NMDA receptors*)* with chronic binge drinking (as compared to water controls). Within the same comparison, we also observed a significant downregulation of genes *Dlg4* (encodes the post synaptic density scaffolding protein, PSD95), *Mir132 (*CREB mediated microRNA*)* and *Actl6a (*encodes the actin-related protein-4 ARP4*)*. These significant changes in expression patterns were absent within the EtOH (CNO) vs. H_2_O (CNO) comparison. In the context of untreated binge drinking, these changes in gene expression likely represent the underlying mechanisms for the increased neuronal complexity observed with chronic binge drinking. Further, the absence of these expression patterns with CNO treatment supports that the expression of these genes is ameliorated with CNO. Support for these mechanisms have also been shown by Uys et al., 2016, where their group observed significant changes in NAc core MSN morphology and expression of PSD95 (*Dlg4*) with chronic intermittent ethanol exposure [42]. Additionally, PSD95 has also been shown to be downregulated by other drugs of abuse such as morphine and cocaine [55,56,57]. 

### 4.3. Effects of Chronic Binge-Drinking and DREADDs on the NAc Transcriptome 

Reductions in ethanol consumption observed during the washout period of the DID paradigm suggest neuroplastic changes as a result of chronic neuronal stimulation in mice expressing hM3Dq. Transcriptomic analyses were carried out to identify differentially expressed genes (DEGs) for each group (Appendix A) and used to identify genes associated with binge-like drinking that were ameliorated by CNO (Appendix A). Using DEG analysis, genes with significantly altered levels of expression were identified and contextualized within biological networks using Gene ontology analysis (GO). Weighted gene covariance network analysis (WGCNA) was used to construct a scale-free network and identify modules. Genes with a high degree of network connectivity (hubs) that correlated with binge-like drinking were used to identify candidate therapeutic targets (Appendix A; additional results and discussion in Appendix A). Gene overlap identified from DE implicated changes in gene expression that were either treatment and/or fluid type specific. The 688 genes changed solely in the group EtOH (VEH) indicated genes whose expression was perturbed exclusively by chronic binge-like alcohol drinking. This is the first time that NAc transcriptomic changes related exclusively to either chronic binge drinking or chronic CNO stimulation of hM3Dq have been identified. For EtOH (CNO), 1431 genes were found to be related exclusively to those genes ameliorated by CNO treatment. Further, 612 genes were identified in the H_2_O(CNO) group and represented genes that were related exclusively to chronic CNO-induced activation of hM3Dq. 

Progression from acute to chronic alcohol intake is thought to incite changes in gene expression levels that occur to maintain system homeostasis [58]. Chronic exposure to alcohol causes dysregulation of these homeostatic mechanisms leading to alcohol abuse and dependence. Several groups have reported on the effects of selection for alcohol behaviors or alcohol drinking on the transcriptome (for various brain regions and strains/species). Mulligan et al. (2011) examined gene expression after binge drinking (immediately after a 4-day DID), and detected a strong relationship between the BEC and a striatal network of gene modules enriched in gene annotations for protein binding and oxidative phosphorylation. Hitzemann et al. (2017) reported a significant overlap between striatal gene expression associated with individual variation (measured 3 weeks after a 4-day DID) and risk for excessive drinking (genes associated with selection as measured in naïve mice) [10]. Many of these genes were related to NMDA-mediated synaptic plasticity. Furthermore, Ferguson et al. (2019) found many changes in brain gene expression between ethanol-naïve HDID-1 and HS/Npt mice, specifically in glutamatergic and GABAergic signaling pathways in the NAc (shell) [59]. The brain regions used in Ferguson et al. (2019) represent circuitry well known to be involved in motivated behaviors, as well as alcohol drinking [59]. *Hdac4* was identified in selection responsive modules as functionally overrepresented. We identified *Hdac4* as a gene of interest in our study and have discussed this below.

We observed 688 genes uniquely differentially expressed in the EtOH (VEH) group, where those exhibiting the most extreme changes in expression were ameliorated in CNO treatment groups. GO analysis of DEGs identified for the EtOH (VEH) group identified the importance of *Hdac4* expression in binge drinking. HDAC4 is a class IIa histone deacetylase that represses transcription when tethered to a promoter (via a multiprotein complex with MEF2C and MEF2D). In pathways such as cellular macromolecule metabolism, positive regulation of cellular processes, negative regulation of biological processes and regulation of macromolecule metabolism, *Hdac4* was consistently found to be overrepresented. Additionally, we observed gene representation related to circadian and neuroimmune pathways, as others have previously reported [8,60,61]. Because *Hdac4* was present in the majority of significant GO pathways, we hypothesized that *Hdac4* played a large role in a number of biological pathways related to chronic binge drinking. We were encouraged to use this approach to test whether pharmacologically targeting HDAC4 could reduce binge-like drinking.

### 4.4. Pharmacologically Targeting HDAC4 to Reduce Binge-Like Drinking

It has been shown that alcohol dependence and withdrawal result in a global reduction of expressed transcripts, accompanied by an increase in expressed transcripts coding for regulators of epigenetic silencing [62]. One such class of enzymes, histone deacetylases (HDAC; transcriptionally silencing enzymes), are inhibited by valproate. Valproate treatment during alcohol withdrawal reduces anxiety in dependent animals [63]. Simon-O’Brien et al. (2012) reported that several classes of HDAC inhibitors and DNA methyltransferase inhibitors decrease alcohol-seeking behaviors [64]. Thus, there is growing evidence that epigenetic modifiers and changes in gene expression play a role in alcohol behaviors. Here, we showed that the chronic treatment with the HDAC4/5 inhibitor LMK 235 reduced binge-like drinking in mice selectively bred to drink to intoxication. Based on these results, we inferred that there were changes in transcriptional mechanisms because we observed that reductions in intake developed over several treatment days. Although *Hdac4* was selected as a target based on transcriptomic data from female mice, the compound reduced ethanol drinking in both male and female mice. Others have shown that HDAC4 is regulated by neuronal activity and plasticity, and inhibiting HDAC4 via LMK 235 can alter neuronal morphology [50,65,66]. Ongoing work is focused on using this data to identify and test compounds with high clinical translational potential.

## 5. Summary

We found chronically increasing NAc activity (4 weeks via CNO/hM3Dq) decreased binge-like drinking. This effect persisted for 7 days post-CNO treatment, suggesting the development of plasticity. We investigated whether this behavioral plasticity could be observed at the morphological and/or transcriptomic levels. We found that chronic binge drinking increased neuronal complexity in the NAc, an effect not present in mice treated with CNO. We detected significant changes in expression of several morphology and plasticity-related genes with binge drinking that were ameliorated with CNO treatment (e.g., *Hdac4*). We tested whether inhibiting HDAC4 would reduce drinking and found that chronic treatment with LMK235, an HDAC4/5 inhibitor, reduced binge-like drinking. Thus, we were able to target specific molecular pathways using pharmacology to mimic the behavioral effects of DREADDs.

## Figures and Tables

**Figure 1 brainsci-10-00109-f001:**
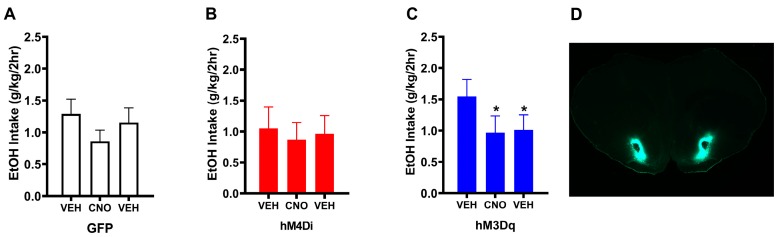
Chronically increasing Gq signaling in the NAc produced lasting reductions in binge-like drinking. Mean (+/- SEM) 2-hr ethanol (EtOH) intake during drinking in the dark (DID) is shown as a function of the three treatment periods for each group as follows: (**A**) control (GFP), (**B**) inhibitory DREADDs (hM4Di), and (**C**) mice expressing excitatory DREADDs (hM3Dq). The first bar is data from week 1 to measure baseline intake in response to vehicle, the second bar is data from weeks 2–5 to measure intake in response to chronic vehicle (VEH) (**A**) or clozapine-N-oxide (CNO) (**B**), and the third bar is data from week 6 to measure intake in response to VEH again (and determine whether there were lasting effects of CNO). (**C**) Significant decreases were observed only in hM3Dq during CNO treatment and washout, relative to baseline intake (one way ANOVA—F(2,35) = 4.84, *p* < 0.05; post-hoc—VEH(baseline) vs. CNO, * *p* < 0.05; VEH (baseline) vs. VEH (washout), * *p* < 0.05). No significant differences in intake over treatment periods were observed for GFP or hM4Di groups. (**D**) 2× image of coronal section showing targeting of NAc core in a representative mouse from this study (where green indicates viral transduction).

**Figure 2 brainsci-10-00109-f002:**
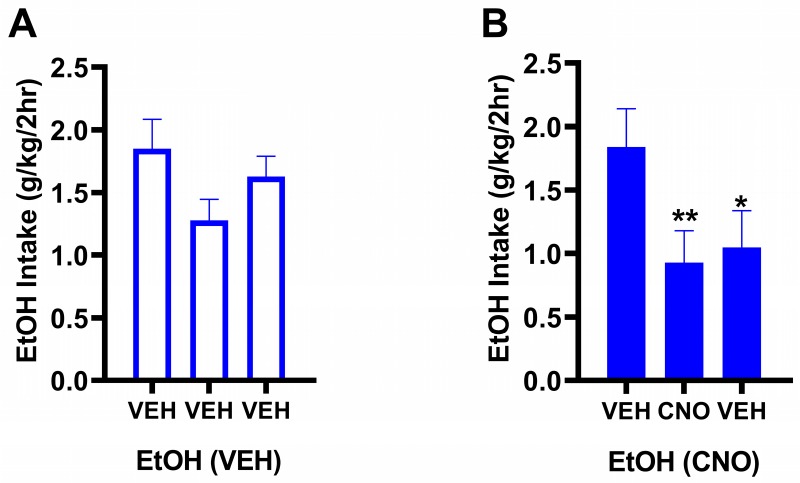
Chronically increasing Gq signaling in the NAc produces lasting reductions in binge-like drinking. Mean (± SEM) 2-h intake during the Drinking in the Dark (DID) paradigm is shown as a function of the three treatment periods. The first bar is data from week 1 to measure baseline intake in response to vehicle, the second bar is data from weeks 2–5 to measure intake in response to chronic VEH (**A**) or CNO (**B**), and the third bar is data from week 6 to measure intake in response to VEH again (and determine whether there were lasting effects of CNO). (**A**) No significant decreases in binge-like EtOH intake were observed in response to chronic treatment with vehicle throughout the duration of the experiment. (**B**) Robust, significant reductions in binge-like EtOH intake were observed in response to CNO treatment and these reductions persisted during the washout period (one way ANOVA—F (2,35) = 10.49, *p* < 0.01; post-hoc test revealed significant differences for VEH baseline vs. CNO (** *p* < 0.01) and VEH baseline vs. VEH washout (* *p* < 0.05)).

**Figure 3 brainsci-10-00109-f003:**
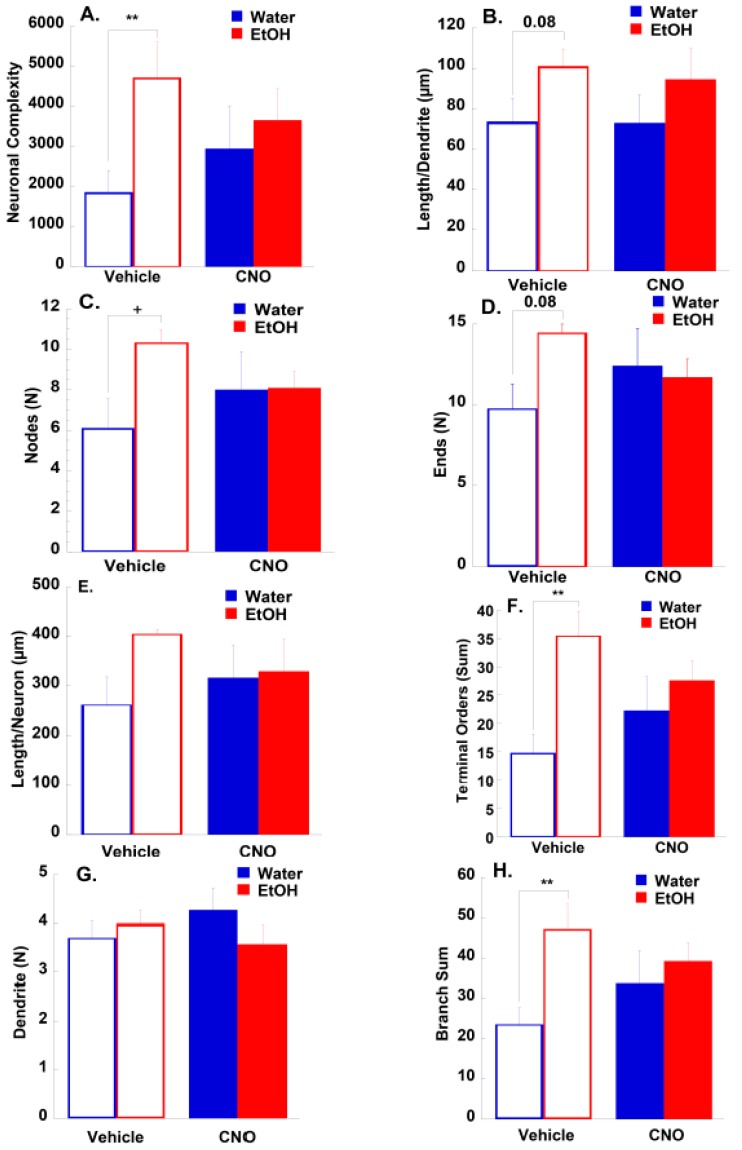
Effects of chronic Gq Designer Receptors Exclusively Activated by Designer Drugs (DREADD) stimulation and drinking on neuronal morphology. Morphometric parameters analyzed for the dendrites of medium spiny neurons. Morphometric parameters of dendrites of nucleus accumbens (NAc) medium spiny neurons of Adult High Drinking in the Dark line 1 (HDID-1) mice injected with either CNO or a vehicle, and exposed to either EtOH or H_2_O were analyzed via Neurolucida Explorer. (**A**) Complexity, calculated via the equation (sum of terminal orders + number of terminals) * (total dendrite length/number of dendrites). (**B**) Average dendrite length (total length/dendrite quantity) in µm. (**C**) Number of nodes per neuron. (**D**) Number of ends per neuron. (**E**) Total dendrite length per neuron, in µm. (**F**) Sum of the terminal orders (the number of sister branches encountered when tracing a dendrite from the tip back to the cell body). (**G**) Number of dendrites per neuron. (**H**) Branch sum per neuron (the total number of segments between nodes, a measure of cell branching). (**I**–**T**) shows Sholl analysis and area under the curve (AUC) of dendrites of nucleus accumbens medium spiny neurons from HDID-1 mice injected with either CNO or a vehicle, and exposed to either EtOH or H_2_O, which were analyzed with Neurolucida Explorer. Radii were 10 µm each and the *x*-axis is “Distance from soma in uM”. Intersections per radius of H_2_O Veh vs. EtOH Veh (**A**), H_2_O Veh vs. EtOH CNO (**B**), and AUC (**C**). Length per radius of H_2_O Veh vs. EtOH Veh (**D**), H_2_O Veh vs. EtOH CNO (**E**), and AUC (**F**). Nodes per radius of H_2_O Veh vs. EtOH Veh (**G**), H_2_O CNO vs. EtOH CNO (**H**) and AUC (**I**). Ends per radius of H_2_O Veh vs. EtOH Veh (**J**), H_2_O CNO vs. EtOH CNO (**K**) and AUC (**L**). Shown in each graph is the mean ± SEM (for neuron number per group see Appendix A). Multilevel analyses were followed with multiple comparison corrections using the Benjamini–Hochberg approach to adjust p-values to the false discovery rate (FDR) (see Table 1). ** *q*-value < 0.01, * *q*-value < 0.05 (after FDR corrections), + = *p*-value < 0.05 (before FDR corrections). All statistics are in Appendix A.

**Figure 4 brainsci-10-00109-f004:**
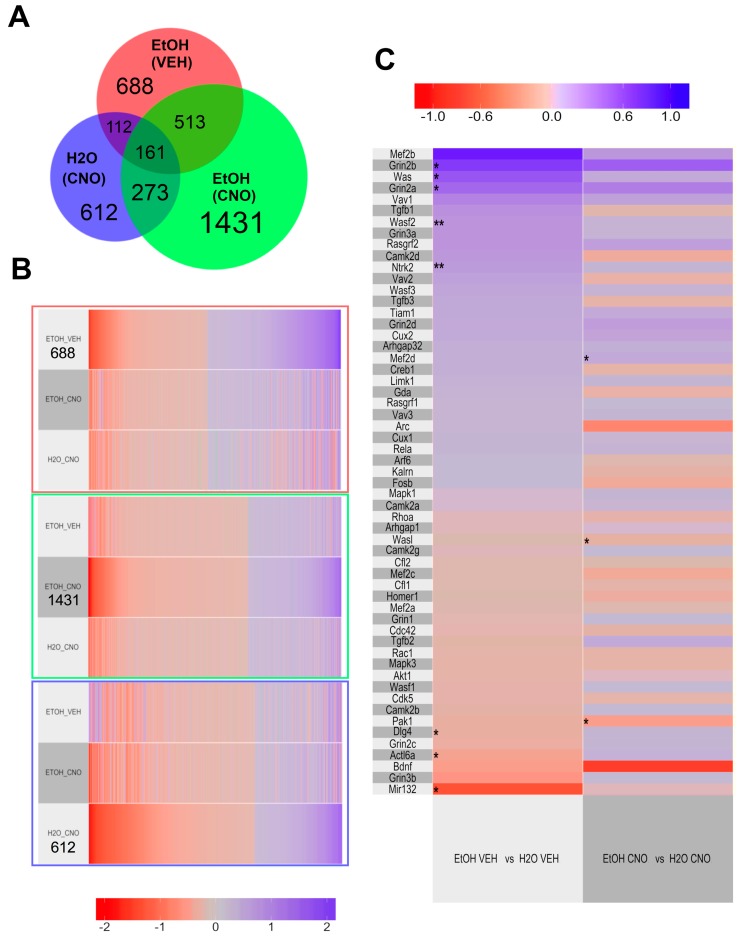
Gene expression changes produced by chronic binge drinking and stimulation of Gq signaling in the NAc. (**A**) The Venn diagram represents overlapping or unique differentially expressed genes (DEGs) (*p* < 0.05) in each group based on a pairwise comparison of all groups to H_2_O(VEH) mice. (**B**) Patterns of DEGs associated with binge-like EtOH drinking identified in (**A**) across groups are visualized by heatmaps, where blue = upregulation, red = downregulation, and gray = minimal to no change in expression. The top panel of (**B**) shows the 688 DEGs associated with EtOH (VEH), the middle panel shows the 1431 associated with EtOH (CNO) and the bottom panel shows the 612 associated with H_2_O (CNO). (**C**) To identify the molecular mechanisms mediating changes in neuronal morphology induced by chronic binge drinking that were ameliorated with CNO, we have highlighted the expression patterns of 57 key genes known to mediate neuronal morphology. DEGs are shown as * *p* < 0.05, ** *p* < 0.01 and the comparisons are labeled within the columns.

**Figure 5 brainsci-10-00109-f005:**
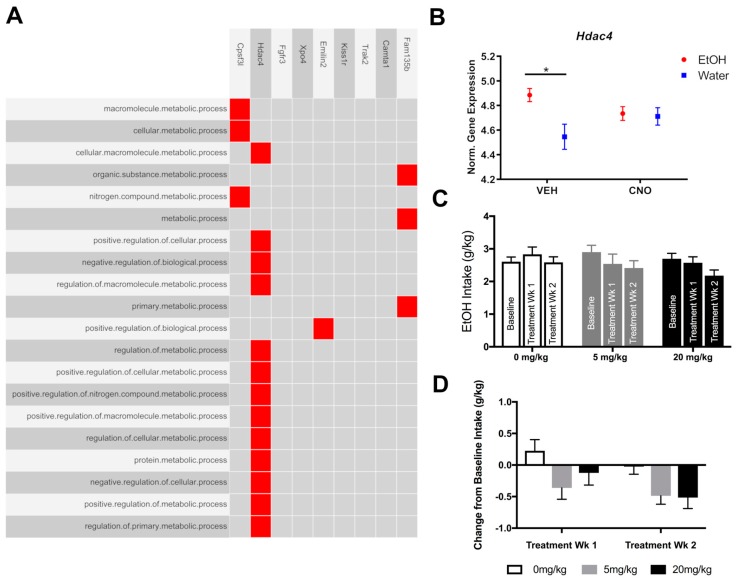
Differential expression of Hdac4 and treatment with HDAC4/5 inhibitor, LMK-235, reduced binge-like alcohol drinking. (**A**) A clustergram is shown to visualize results from GO performed on ranked DEGs identified from EtOH(VEH). Enriched terms are listed in rows and the corresponding genes ranked first in each term are shown in each column. Red indicates a gene is ranked first within a term. Grey indicates a gene is not ranked first. (**B**) Limma normalized gene expression levels of *Hdac4* are shown for each treatment and fluid group. A significant increase in normalized expression was observed between H_2_O and EtOH groups receiving vehicle treatment (* *p* < 0.05). (**C**) Average weekly ethanol intake after treatment with 0, 5 or 20 mg/kg LMK-235. Two-way ANOVA revealed a significant main effect of treatment period on ethanol intake (F(2,152) = 5.16, *p* < 0.01). (**D**) Ethanol intake presented as the average difference score from baseline intake for each treatment week. Here, significant reductions were observed for both doses over a two-week period (Two-way ANOVA (dose × treatment week) revealed a significant main effect of dose and a significant main effect of treatment week (dose: F(2,76) = 3.60, *p* < 0.05; treatment week: F(1,76) = 2.44, *p* < 0.05)).

**Table 1 brainsci-10-00109-t001:** The number of animals and cells per group that were included in morphological analyses are provided here.

Number of Animals and Cells Per Group Included for Morphological Analyses							
Group	Animal (*n*)	Cells/Brain#1	Cells/Brain#2	Cells/Brain#3	Cells/Brain#4	Cells/Brain#5	Total Cells
H_2_O+Veh	4	9	5	6	9	0	29
H_2_O+CNO	5	5	7	5	6	13	36
EtOH+Veh	4	6	10	5	8	0	29
EtOH+CNO	4	6	5	6	6	0	23
Totals	17						117

*n* = number of animals, H_2_O = Water, Veh = Vehicle, CNO = clozapine-n-oxide, EtOH = ethanol.

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
