# Peer review of "Chronic Chemogenetic Stimulation of the Nucleus Accumbens Produces Lasting Reductions in Binge Drinking and Ameliorates Alcohol-Related Morphological and Transcriptional Changes"

_brainsci, 2020, doi:10.3390/brainsci10020109_

Round 1

Reviewer 1 Report

The paper represents a thorough investigation of the relation between accumbal stimulation and alcohol consumption and adds a great deal to the alcohol literature.  

-----------------------------------------

The authors present a high quality work in a subject of high scientific relevance for neuroscience. In spite of these qualities, the submission package is missing the proper supplementary files, which precludes a proper review.The missing files are essential for the proper comprehension and critical analysis of the paper.

I suggest the authors to resubmit the paper with the proper supplementary files for a second round of revision.

Author Response

We are pleased to submit the revised version of our manuscript. The blue text in the revised manuscript represents revisions in response to the reviewers’ comments. We appreciate the constructive criticisms of the reviewers and have addressed each of their concerns as outlined below.

We thank the reviewers and Editor for their time and careful consideration. The manuscript has been greatly improved by addressing these concerns and comments.

As instructed, we disregard Reviewer 1’s comment regarding missing supplementary files.

Reviewer 2 Report

General comment: 

In this manuscript, using HDID-1 mice as an animal model of binge alcohol drinking, the authors have attempted to study how the chemogenetic stimulation of the NAc ameliorates the alcohol related morphological and transcriptional changes.  Overall, the manuscript is good and well written, with enough background information given for performing the research. The results from the study provide insight into the alterations  at the structural level as well as transcriptome level of the neurons of the NAc following binge alcohol drinking. However,  the manuscript lacks clarity in the methodology. Also, it would be better if the authors can write a brief conclusion of their study.

Specific comments: 

The only drawback of the manuscript is limited information in the materials and methods section.I would suggest the authors to provide the basic information of their experiments in the materials and methods sections as well rather than putting all the information in the supplementary information.

Comments regarding Materials and Methods:

In experiment 1 and 2,only mice is mentioned, whereas in experiment 3, HDID-1 mice is mentioned.

There is no mention of the age and sex of the mice.

No mention of drugs in Drugs subsection.

Line 101-102: This part is very confusing. When I refer forward to line 185-186, authors have mentioned 4 groups. Its better to clarify and write 4 groups and the regimen of intervention in each groups separately.

Microscopy and RNA sequencing: It would be better just to put basic information in these sections, so that the readers can follow through the manuscript.

Regarding experiment 3, just for my curiosity, can it be done following the protocol of experiment 1 and 2?

Other minor comments;

The full form of the abbreviated terms (GPCR, CIE, etc) should me mentioned at least once in the manuscript.

Line 29-31 : no reference is given

Structuring the results section with subheadings will give further clarity to the findings  of the study.  

Line-247-248 : “Modest, but significant”: this sounds contradictory.

Line 306: full name of CIE

Line 366: which homeostasis are the authors referring to should be mentioned.

Line 389-390: This sentence lacks clarity with no clear function of Hdac4 mentioned.

Line 435 (Figure 2): The authors have not mentioned what the X-axis represents in both the bar diagrms.(VEH-VEH-VEH and VEH-CNO-VEH). Do they represent the same time points?

Line 444 (Figure 3): What is the x-axis of the graphs?

Line 487-488: There is no figure 7A in the manuscript.

Author Response

Reviewer 2 comments and author responses (preceded by **):

**We are pleased to submit the revised version of our manuscript. The blue text in the revised manuscript file denotes revisions in response to the reviewers’ comments. We appreciate the constructive criticisms of the reviewers and have addressed each of their concerns as outlined below. We thank the reviewers and Editor for their time and careful consideration. The manuscript has been greatly improved by addressing these concerns and comments.

General reviewer 2 comment:

In this manuscript, using HDID-1 mice as an animal model of binge alcohol drinking, the authors have attempted to study how the chemogenetic stimulation of the NAc ameliorates the alcohol related morphological and transcriptional changes. Overall, the manuscript is good and well written, with enough background information given for performing the research. The results from the study provide insight into the alterations at the structural level as well as transcriptome level of the neurons of the NAc following binge alcohol drinking.

However, the manuscript lacks clarity in the methodology. Also, it would be better if the authors can write a brief conclusion of their study.

**We incorporate additional methodology details into the main manuscript and added a brief summary at the end of the manuscript.

Specific comments:

The only drawback of the manuscript is limited information in the materials and methods section. I would suggest the authors to provide the basic information of their experiments in the materials and methods sections as well rather than putting all the information in the supplementary information.

**We now incorporate all methodology details from the supplement into the main manuscript.

Comments regarding Materials and Methods:

In experiment 1 and 2, only mice is mentioned, whereas in experiment 3, HDID-1 mice is mentioned.

**HDID-1 mice were used for all experiments. Additional text has been added to various sections to clarify.

There is no mention of the age and sex of the mice.

**Additional text has been added to various sections to clarify.

No mention of drugs in Drugs subsection.

**Additional text has been added to the Drugs subsection.

Line 101-102: This part is very confusing. When I refer forward to line 185-186, authors have mentioned 4 groups. Its better to clarify and write 4 groups and the regimen of intervention in each groups separately.

** Additional text has been added to clarify.

Microscopy and RNA sequencing: It would be better just to put basic information in these sections, so that the readers can follow through the manuscript.

**This information has been added to the main manuscript.

Regarding experiment 3, just for my curiosity, can it be done following the protocol of experiment 1 and 2?

**It could be done in this way. However, we found no extant data in the literature on the effects of testing past 8 treatments and we wanted to be able to compare our behavioral results with published results.

Other minor comments;

The full form of the abbreviated terms (GPCR, CIE, etc) should me mentioned at least once in the manuscript.

**GPCR and CIE have been defined at first mention.

Line 29-31 : no reference is given

**References have been added.

Structuring the results section with subheadings will give further clarity to the findings of the study.

**Subheadings have been added and underlined.

Line-247-248 : “Modest, but significant”: this sounds contradictory.

**Modest has been removed.

Line 306: full name of CIE

**CIE has been defined.

Line 366: which homeostasis are the authors referring to should be mentioned.

**This sentence has been edited to clarify.

Line 389-390: This sentence lacks clarity with no clear function of Hdac4 mentioned.

**We edited the sentence for clarity and describe HDAC4.

Line 435 (Figure 2): The authors have not mentioned what the X-axis represents in both the bar diagrms.(VEH-VEH-VEH and VEH-CNO-VEH). Do they represent the same time points?

**The caption for Figures 1 and 2 have been edited to clarify that each bar represents a treatment period and additional text has been added to describe what each bar represents with respect to the experimental timeline.

Line 444 (Figure 3): What is the x-axis of the graphs?

**Distance from soma in uM. Additional text has been added to the caption to clarify.

Line 487-488: There is no figure 7A in the manuscript.

**We apologize for the error. Reference to Fig. 7a has been removed.

Reviewer 3 Report

This study used mice bred for binge-like drinking and DREADD manipulation of medium spiny neurons of the nucleus accumbens (NAc). DREADD activation of GPCRs (via Gq/CNO/hM3Dq) in the NAc reduced binge-like drinking. Additional experiments determined that decreased drinking was mediated by reducing “neuronal complexity” in the NAc via plasticity related genes such as Hdac4 and that pharmacological manipulation of Hdac4 similarly reduced drinking.

This is a timely and well-considered work. A number of experiments – including behavior, cellular morphology, and gene analysis - were conducted and all were appropriate, but the presentation of the methods was hard to follow. Presentation of results was fine.

A revision to clarify some points as indicated below should make the manuscript better suited for publication.

Not clear what is meant by “neuronal complexity” in the abstract. While the concept is complex and later introduced, a sentence describing this term is warranted in the abstract.

Please define “chronic” drinking and “chronic” DREADD stimulation in abstract and introduction.

Was the AAV surgery conducted prior to exposure to DID? Timing not clear.

Not clear the purpose of the different viruses used: e.g., rAAV2-hSyn-eGFP = control? rAAV2-hSyn-HA-hM3D(Gq)-IRES-mCitrine = excitatory; rAAV2-hSyn-HA-hM4D(Gi)-IRES-mCitrine = inhibitory

Also not clear the goals of the 3 different experiments.

Were BALs measured?

Results, experiment 2: much of the information presented in the Results section belongs in the Methods section.

It seems that measures of “neuronal complexity” were quantified because there were no significant changes in the more typical measures of total dendritic length or number of dendrites. Given that there are few publications using the concept of “neuronal complexity” as presented (pubmed returns 12 manuscripts), what additional evidence and figures (photos?) can the authors provide to support the rigorous assessment of these measures? Indeed, in the Wang 2015 paper, how measures of “complexity” were derived are clearly shown in Fig 5 including a sample image of biocytin-filled D1R MSN and its dendritic branches. Furthermore, it is not clear from whence the equation: (sum of terminal orders + number of terminals) * (total dendritic length/number of dendrites) was derived. Could not find the appropriate reference. If this is a measure automatically derived from Neurolucida, then this needs to be clarified. More generally, these issues are of concern since manual tracing methods are susceptible to observer bias and variability (Srinivasan 2020, J NS Methods).

Justification for only using female mice?

In Fig 1. Why was binge drinking highest at baseline in the hM3Dq mice? (1.5g/kg vs. 1g/kg in the hM4Di). This baseline difference in drinking and not the effects of CNO could account for statistical significance of CNO stimulation (i.e., if baseline drinking in hM2Dq mice was only 1g/kg, chronic CNO stimulation would not show a statistically effect on EtOH consumption).

Both in Fig 1A and Fig 2A, the pattern suggests reduced drinking (though perhaps not statistically significant) during week 2 [in 1A in response to CNO and 2A in response to Vehicle]. Please comment.

In Figs 3I-T what is the Y axis (i.e., 0-140 of what???)

In Fig 3I, J, L, what is happening in controls at the 120 point? Why is there a sudden and transient elevation in the variable of interest?

Fig 5b – not clear. Did treatment with LMK-235 reduce drinking? At week 2 of 20mg/kg? if so, please indicate with asterisk.

Define GPCRs.

Please be more explicit: Cassataro et al. (2014) found that CNO reduced binge-like drinking in C57BL/6J males (Cassataro et al., 2014). With what kind of DREADDs had they infected mice?

Regarding this statement: Interestingly and unexpectedly, no effect of CNO was observed in water drinking mice, suggesting that chronically increasing Gq signaling in the NAc does not produce robust and lasting changes in neuronal complexity. Wouldn’t chronic CNO stimulation in non-EtOH exposed mice nevertheless cause a change in the same direction? i.e., to REDUCE complexity? Why would chronic activation of Gq in EtOH0-naive animals be expected to increase neuronal complexity?

Please comment: Do the authors think that they did not observe an increase in dendritic length or number of dendrites as previously observed (DePoy 13, Wang 15, Uys 16) because of the different EtOH exposure paradigms?

Please include a concluding paragraph summarizing results and highlighting significance.

Author Response

Reviewer 3 comments and author responses (preceded by **):

**We are pleased to submit the revised version of our manuscript. The blue text in the revised manuscript file denotes revisions in response to the reviewers’ comments. We appreciate the constructive criticisms of the reviewers and have addressed each of their concerns as outlined below. We thank the reviewers and Editor for their time and careful consideration. The manuscript has been greatly improved by addressing these concerns and comments.

This study used mice bred for binge-like drinking and DREADD manipulation of medium spiny neurons of the nucleus accumbens (NAc). DREADD activation of GPCRs (via Gq/CNO/hM3Dq) in the NAc reduced binge-like drinking. Additional experiments determined that decreased drinking was mediated by reducing “neuronal complexity” in the NAc via plasticity related genes such as Hdac4 and that pharmacological manipulation of Hdac4 similarly reduced drinking.

This is a timely and well-considered work. A number of experiments – including behavior, cellular morphology, and gene analysis - were conducted and all were appropriate, but the presentation of the methods was hard to follow. Presentation of results was fine.

A revision to clarify some points as indicated below should make the manuscript better suited for publication.

Not clear what is meant by “neuronal complexity” in the abstract. While the concept is complex and later introduced, a sentence describing this term is warranted in the abstract.

**We edited the sentence to describe that neuronal morphology is altered. Of course, we prefer to be more specific. However, space in the abstract far too limited to expand on this complex concept.

Please define “chronic” drinking and “chronic” DREADD stimulation in abstract and introduction.

**We noted the amount of time we tested for each “chronic” reference in the abstract and introduction.

Was the AAV surgery conducted prior to exposure to DID? Timing not clear.

**Yes. To clarify the timeline, additional details have been added to the methods sections for Experiment 1 and 2.

Not clear the purpose of the different viruses used: e.g., rAAV2-hSyn-eGFP = control? rAAV2-hSyn-HA-hM3D(Gq)-IRES-mCitrine = excitatory; rAAV2-hSyn-HA-hM4D(Gi)-IRES-mCitrine = inhibitory

**We now explain this in the introduction and in the methods.

Also not clear the goals of the 3 different experiments.

**The goals of each experiment are discussed in the introduction and prior to the results for each experiment in the results section.

Were BALs measured?

**No. In our experience, collecting blood for BAL determination results in reduced ethanol intake on subsequent days. These were all chronic experiments and we could not identify a time when this effect would not be a confound for interpretation of the data. We believe this would be a confounding factor for all experiments - especially for the transcriptomics and morphology experiments, where there were water groups (collecting blood samples from water groups is not advisable).

Results, experiment 2: much of the information presented in the Results section belongs in the Methods section.

**It is unclear what edits are requested. One interpretation is that the reviewer prefers the purpose statements (that preface each of the results) be moved to the methods section. We can find no guiding statements from the journal here, and thus, elect to keep these purpose statements in the results section. If we are guessing incorrectly or if the editor and reviewer find this unacceptable for publication, we will need additional guidance on what should be moved to the methods section.

8a. It seems that measures of “neuronal complexity” were quantified because there were no significant changes in the more typical measures of total dendritic length or number of dendrites.

**I assure the reviewer this is not the case. This measure was deemed of interest prior to data analysis.

8b. Given that there are few publications using the concept of “neuronal complexity” as presented (pubmed returns 12 manuscripts), what additional evidence and figures (photos?) can the authors provide to support the rigorous assessment of these measures? Indeed, in the Wang 2015 paper, how measures of “complexity” were derived are clearly shown in Fig 5 including a sample image of biocytin-filled D1R MSN and its dendritic branches.

**Our co-authors and collaborators, Drs. Goeke and Guizzetti, have extensive experience measuring the effects of ethanol on morphology and suggested incorporating the neuronal complexity measure (automatically calculated by neurolucida during tracing) because it permits comparison of fundamentally different cell type and thus, may have broader implications for future work. They have previously published that ethanol increased apical dendrite complexity in male and female pups neonatally exposed to ethanol (Neuroscience 2018. 374:13-24). Additionally, a search in Google Scholar revealed numerous peer reviewed published manuscripts have used this measure. Color neurolucida traces accompany our raw data files and representative traces for neurons from each group can be supplied if required for publication.

8c. Furthermore, it is not clear from whence the equation: (sum of terminal orders + number of terminals) * (total dendritic length/number of dendrites) was derived. Could not find the appropriate reference. If this is a measure automatically derived from Neurolucida, then this needs to be clarified.

**Added to the main text methods subsection: Neuronal complexity is a measure automatically derived from Neurolucida.

8d. More generally, these issues are of concern since manual tracing methods are susceptible to observer bias and variability (Srinivasan 2020, J NS Methods).

**Indeed, we are also concerned with bias and replication. To reduce bias, the researcher (co-author, Evan Firsick) that performed the traces was blind to treatment group (we used random number generator to code slides for tracing) and analyses were carried out by a separate individual (co-author, Calla Goeke) that had no knowledge of treatment groups. To assess variability, Evan Firsick re-traced the first quarter of samples initially traced in order to determine whether the data for those samples were replicable (indeed, they were).

Justification for only using female mice?

**Additional text has been added to the methods subsection for experimental animals.

In Fig 1. Why was binge drinking highest at baseline in the hM3Dq mice? (1.5g/kg vs. 1g/kg in the hM4Di). This baseline difference in drinking and not the effects of CNO could account for statistical significance of CNO stimulation (i.e., if baseline drinking in hM2Dq mice was only 1g/kg, chronic CNO stimulation would not show a statistically effect on EtOH consumption).

** A caveat of this work is that we were unable to pseudorandomize animals to groups as all mice underwent surgery and were injected with different AAVs (hM3Dq, hM4Di, or GFP). This can sometimes lead to variable EtOH intake between groups and experiments. However, only mice with confirmed placements were included in the analyses. Therefore, we could not a priori pseudorandomize individuals to specific groups prior to the beginning of treatment to minimize group differences, which can lead to group differences in EtOH intake (at baseline). Conducting these studies with sub-region specific injections of an AAV into the NAc core, especially in mice, is very challenging. Including vehicle and CNO administration for each animal (within subject analysis), as well as always including an AAV-GFP group allows us to focus on changes within groups in response to treatment. We find that CNO/hM3Dq in the NAc reduces binge-like drinking in 3 previously published experiments in female C57BL/6J mice (Purohit et al., 2018) and now here in 2 different experiments carried out in HDID-1 female mice. Together, this gives us confidence that our studies using CNO/hM3Dq in the NAc are replicable and generalize to more than one strain of mice.

Both in Fig 1A and Fig 2A, the pattern suggests reduced drinking (though perhaps not statistically significant) during week 2 [in 1A in response to CNO and 2A in response to Vehicle]. Please comment.

** We acknowledge the reviewer’s comment, but due to a lack of “trending” or actual statistical significance for data presented in Figure 1a and 2a, we decline to offer speculative comments in the manuscript. See comment above for comment on variability in data.

In Figs 3I-T what is the Y axis (i.e., 0-140 of what???)

**Distance from soma in uM. Additional text has been added to the caption to clarify.

In Fig 3I, J, L, what is happening in controls at the 120 point? Why is there a sudden and transient elevation in the variable of interest?

**There appears to be elevated variability in the data in the number of intersections at that particular radial interval (120 uM from the soma).

Fig 5b – not clear. Did treatment with LMK-235 reduce drinking? At week 2 of 20mg/kg? if so, please indicate with asterisk.

**Due to the lack of a significant dose x treatment period interaction for data presented in figure 5b, we did not pursue post-hoc multiple comparison testing for the week two 20mg/kg data.

Define GPCRs.

**It has been defined.

Please be more explicit: Cassataro et al. (2014) found that CNO reduced binge-like drinking in C57BL/6J males (Cassataro et al., 2014). With what kind of DREADDs had they infected mice?

**Additional text has been added to indicate they saw reduced intake with CNO when hM4Di was expressed in the NAc.

Regarding this statement: Interestingly and unexpectedly, no effect of CNO was observed in water drinking mice, suggesting that chronically increasing Gq signaling in the NAc does not produce robust and lasting changes in neuronal complexity. Wouldn’t chronic CNO stimulation in non-EtOH exposed mice nevertheless cause a change in the same direction? i.e., to REDUCE complexity? Why would chronic activation of Gq in EtOH0-naive animals be expected to increase neuronal complexity?

**We agree. Directionality is not indicated in our statement. We thought we might see an effect of CNO/hM3Dq in water drinking animals, but we did not.

Please comment: Do the authors think that they did not observe an increase in dendritic length or number of dendrites as previously observed (DePoy 13, Wang 15, Uys 16) because of the different EtOH exposure paradigms?

** It is not clear whether strain, sex, or drinking paradigm (or some combination of these factors) contributes to these differences.

Please include a concluding paragraph summarizing results and highlighting significance.

**A summary has been added.